# Open-Heart Cardio-Thoracic Biological Valve Replacement Following Complicated Transcatheter Aortic Valve Implantation

**DOI:** 10.3390/jpm13050838

**Published:** 2023-05-16

**Authors:** Aneta Klotzka, Patrycja Woźniak, Marcin Misterski, Michał Rodzki, Mateusz Puślecki, Marek Jemielity, Marek Grygier, Aleksander Araszkiewicz, Sylwia Iwańczyk, Piotr Buczkowski

**Affiliations:** 1Department of Cardiology, Poznan University of Medical Sciences, Długa 1/2 Street, 61-848 Poznan, Poland; 2Cardiac Surgery and Transplanthology Department, Poznan University of Medical Sciences, Długa ½ Street, 61-848 Poznan, Poland

**Keywords:** aortic stenosis, transcatheter aortic valve implantation, TAVR, paravalvular leak

## Abstract

Transcatheter aortic valve implantation (TAVI) is currently becoming the method of choice in high-risk patients with severe aortic valve stenosis. Post-TAVI complications are more common owing to the increasing use of the method. The majority of TAVI complications derive from concomitant aortic stenosis with moderate/severe aortic insufficiency, paravalvular leak, and atrioventricular block. The contemporary TAVI qualification process includes a thorough echocardiography and angio-CT of the aorta, which is crucial in assessing valve measurements, determining the position of the coronary arteries branching from the aorta, and choosing the optimal valve size. We present the case report of an 81-year-old patient admitted to our hospital because of exacerbation of the clinical condition and development of pulmonary edema a few days after TAVI. Despite the reduction of the initial leak, an echocardiographic examination revealed the remaining severe paravalvular aortic leakage. We performed open-heart cardio-thoracic surgery, explanted the TAVI valve, and implanted the biological prosthesis (Edwards Perimount Magna size 25). Introduction of new interventional treatment approaches and the availability of imaging tools have substantially reduced the incidence of significant paravalvular leak and offered a better prognosis for patients undergoing TAVI.

## 1. Introduction

Aortic stenosis (AS) is one of the most common aortic valve diseases. The sclerotic process is caused by degeneration and calcification of the cusps and/or annulus of the bicuspid or normal trileaflet aortic valve [1]. This degenerative process tends to progress with advancing age [2], which is associated with an increased risk of myocardial infarction (MI) and overall cardiovascular and all-cause mortality [3].

The only definitive treatment for severe AS is surgical or transcatheter aortic valve replacement (TAVR). Although surgical aortic valve replacement (SAVR) is the gold standard [4], transcatheter aortic valve implantation (TAVI) is steadily becoming the method of choice in high-risk patients with severe aortic valve stenosis [5]. Post-TAVI complications are more common owing to the increasing use of the method. The majority of TAVI complications derive from concomitant aortic stenosis (SA) with moderate/severe aortic insufficiency (IA), paravalvular leak (PVL), and atrioventricular (AV block).

Echocardiography and angio-CT are crucial in precisely assessing valve measurements and choosing the optimal valve size.

## 2. Case Report

An 81-year-old patient was admitted to our hospital with a clinical picture of pulmonary edema. He suffered from a complex aortic valve defect with a predominance of severe aortic stenosis.

On admission, he was complaining about shortness of breath and impaired tolerance of exertion for several days. In addition, his past medical history included hypertension and prostate cancer (diagnosed three years ago, did not agree to radiation therapy). His medications included: acetylsalicylate, statin, beta-blocker, torasemide, furosemide, alpha-blockers, and steroid 5-α-reductase inhibitor type II.

The echocardiography revealed the calcified aortic valve manifesting with Aortic Valve Peak Gradient (PGmax) 85 mmHg and Aortic Mean Gradient (Pgmean) 49 mmHg, AVA (Aortic Valve Area) 0.9 cm^2^. LVEF was 50%. No significant lesions were described in the coronary angiography. The CT revealed that the dimension of the annulus pre-TAVI was 24 mm, while the bulb was 34 mm (Figure 1).

After heart team consultation, the patient was disqualified from the surgical procedure because of his concomitant diseases and high surgical risk. He was qualified for the transcatheter aortic valve implantation (the TAVI method) within the TAVI Clinic. The EuroSCORE II result was 9.44%.

The patients are not usually intubated during the TAVI procedure. They are, however, under general anesthesia—that is, they are sedated, but they do not require muscle relaxants, so they do not need to be intubated. The procedure was performed under Monitored Anesthetic Care (MAC).

We punctured the right and left femoral arteries and introduced the vascular sheath. Then, we inserted two Proglide systems and an 18F sheath to introduce a temporary pacing electrode. After the evaluation of the calcified aortic valve and transvalvular gradient, we inserted the 26 mm Medtronic CoreValve Evolut Pro + valve under angiography. The maneuver required six repositions. After relatively deep valve implantation, the aortic prosthesis shifted toward the left ventricle by 6–8 mm after release from the implantation system. We observed severe perivalvular aortic regurgitation—we post-dilated the valve with a Nucleus ZX-Med balloon (26 mm × 4.0). We achieved better adherence to the valve scaffolding and achieved a significant reduction of the PVL. Hemodynamic parameters did not present a significant aortic insufficiency. We reduced aortic regurgitation from severe to moderate with diastolic RR in the aorta of 54 mmHg and end-diastolic pressure in the left ventricle of 18 mmHg. Both femoral arteries were closed with the Angioseal system. Because of the perioperative left bundle branch block and the deep implantation of the prosthesis, we did not remove the temporary pacing electrode. The hemodynamically stable patient was transferred to the Cardiac Intensive Care Unit (ICU).

The sedated patient remained under observation for several hours in the Cardiac ICU. The electrolyte balance was restored. There was no post-procedural bleeding. On the same day, in the evening hours, the sedation was steadily withdrawn.

Despite the reduction of the initial leak, which improved the diastolic left ventricle pressure, the post-procedural ECHO revealed the remaining mild paravalvular aortic leakage. No pathological amount of pericardial fluid was detected. Significant concentric LV wall hypertrophy was observed with concomitant disturbance in ventricular relaxation. The further hospital stay was uncomplicated, and the patient was discharged home on the 13th day of hospitalization.

However, a few days after discharge, the patient was readmitted to the clinic because of the exacerbation of the clinical condition and development of pulmonary edema (Figure 2). Control ECHO revealed severe paravalvular aortic leakage (Figure 3). The heart team debated whether the patient would benefit more from a conventional surgical approach rather than the ViV-TAVI procedure. Following the thorough heart team consult, the cardio-thoracic surgery was performed in January 2023.

The surgery was performed under general anesthesia, and the patient was intubated. Our team performed the midline sternotomy. Despite good contractility of both ventricles, left ventricular hypertrophy was observed.

We inserted a cannula into the left femoral artery and two venous cannulas into the superior vena cava (SVC) and the inferior vena cava (IVC) for separable venous flow with subsequent initiation of extracorporeal circulation (ECC). Cardioplegia solution is high in potassium and low in sodium. Its administration causes significant blood electrolyte disturbances (including central pontine myelinolysis in extreme cases). Therefore, if feasible, we usually cannulate the SVC and IVC separately, hence isolating the venous circulation by SVC and IVC clamp. It enables us to the administer cardioplegia to the right atrium by a small incision. After passing through the entire coronary vascular system, the cardioplegia solution goes back to this right atrium, and we remove it with an external suction tube.

In the next step, the heart was fibrillated, and the ascending aorta was cross clamped. Following the aortotomy, the proximal part of the stent of the previously implanted Evolut valve spontaneously migrated into the lumen of the left ventricle. It partially rested upon the anterior leaflet of the mitral valve.

Subsequently, the frozen 0.9% solution of natrium chloride was placed into the lumen of the implanted valve to achieve plasticization of the nitrile stent, which facilitated the explantation of the valve (Figure 4), and the biological prosthesis (Edwards Perimount Magna size 25) was implanted (Figure 5). The heart was successfully defibrillated, with a recurrence of sinus rhythm. After the appropriate reperfusion time, the CPB was terminated. Without complications, the chest was closed using steel wires and sutures on subcutaneous tissue and skin.

Following the surgery, the sedated patient remained under observation for several hours in the Cardiac ICU. The patient was re-warmed, and electrolyte balance was restored. There was no post-procedural bleeding. On the same day, in the evening hours, the sedation was steadily withdrawn, and the patient was extubated without any complications. He was discharged home two weeks later.

## 3. Discussion

The patient’s condition deteriorated because of the increasing PVL and the constriction of the mitral valve leaflet. Therefore, it was decided to perform the conventional open-heart surgery, alternatively to ViV-TAVI [6].

Choosing the right device size to provide optimal positioning is feasible after pre-procedural imaging, including multi-slice computed tomography (MSCT), echocardiography MSCT, hemodynamic indices, and 3-dimensional angiographic reconstruction by rotational aortic root angiogram before and in the course of the procedure [7]

Several interventional alternatives can reduce the degree of PVL. They include balloons, snares, and valve-in-valve (ViV) [8].

PVL is a common complication post-TAVI, though the impact and etiology of the degree of post-TAVI PVL and mortality require further investigation. It is, however, clear that it impacts both short- and long-term survival negatively [9]. There are several factors associated with an increased risk of PVL. Hagar et al. concluded that ≥mild PVL after TAVI is common and can be predicted by aortic root calcification volume, larger annulus dimensions, and pre-TAVI transvalvular peak velocity, with calcification volume being an independent predictor for PVL [10]. Furthermore, their research also reported that annulus ellipticity, left ventricular outflow tract nontubularity, and diameter-derived prosthesis mismatch apparently have no role in predicting PVL. In contrast, leaflet body calcifications and cusp calcifications have positive predictive value (PPV) in predicting PVL. The anatomy of the aortic annulus, which is the device landing zone (DLZ) for both self-expanding and balloon-expandable aortic valve prostheses, plays an important role in the etiopathogenesis of PVL. Researchers described no association between ≥mild PVL and increased risk of all-cause and cardiovascular mortality at 1-year follow-up [11].

According to Pibarot et al., prosthesis–patient mismatch (PPM) incidence is lower after TAVR when compared to SAVR in patients with severe AS. Patients with PPM after SAVR have worse survival and less LV mass regression than those without PPM. TAVR may be preferable to SAVR in patients with a small aortic annulus susceptible to PPM to facilitate LV mass regression and reduce postoperative mortality [12]. 

In their research, Musallam et al. demonstrated the correlation between the presence of larger calcifications and at least moderate aortic regurgitation (AR) [13].

Nonetheless, Ewe et al. discovered that in using echocardiographic evaluation, PVL is most likely to occur if there is a large volume of calcium at the wall of the valve cusp. At the same time, the risk is lower if the calcifications are located on the free cusp margins or within the valve cusps [14].

The data suggest that the improper depth of an implanted valve is linked to an increased risk of PVL. Until recently, it was impossible to reposition the prosthesis before the final release when the effect of the implanted valve was suboptimal Therefore, it is necessary to employ imaging techniques such as transthoracic echocardiography (TTE), transesophageal echocardiography (TEE), multirow-detector computed tomography (MDCT), aortography, or magnetic resonance [15]. 

Furthermore, Kumar et al. investigated the utility of a time-integrated aortic regurgitation index (TIARI) calculated immediately after valve deployment. They demonstrated that the method can be valuable in predicting balloon postdilation among patients undergoing TAVR. They concluded that the lower the residual TIARI after TAVR, the higher patient mortality rates [16].

In their research, Hovasse et al. found that a snare loop-assisted device minimizes the occurrence of PVL [17].

Acute leaks can be managed with repeated balloon post-dilation (PD) of the under-expanded valve to optimize the expansion and better engagement of the device with native valve anatomy [18].

Implanting a second valve may be beneficial when other techniques fail to manage PVL. In the Italian registry of 663 patients, 3.6% of patients had a ViV procedure. The outcome was similar regarding safety and efficacy for those with ViV and those with a single valve at 1-year follow-up [19].

Al-Abcha et al. conducted a meta-analysis comparing the safety and efficacy of ViV-TAVI and redo-SAVR in failed bioprosthetic valves. They concluded that both techniques are associated with a similar risk of all-cause mortality, cardiovascular mortality, myocardial infarction, permanent pacemaker implantation, and the rate of PVL. In the ViV-TAVI group, the stroke, major bleeding, and procedural and 30-day mortality rates were significantly lower compared with the redo-SAVR group [20].

## 4. Conclusions

To conclude, the introduction of new interventional treatment approaches and the availability of imaging tools enable a substantial reduction in the incidence of significant PVL and a better prognosis for patients undergoing TAVI.

The relatively high position of the TAVI scaffold poses a risk of the lack of space left for a cannula or clamp on the ascending aorta. Hence, sometimes it is preferable to cannulate the aorta from underneath the aortic valve via femoral artery cannulation.

## Figures and Tables

**Figure 1 jpm-13-00838-f001:**
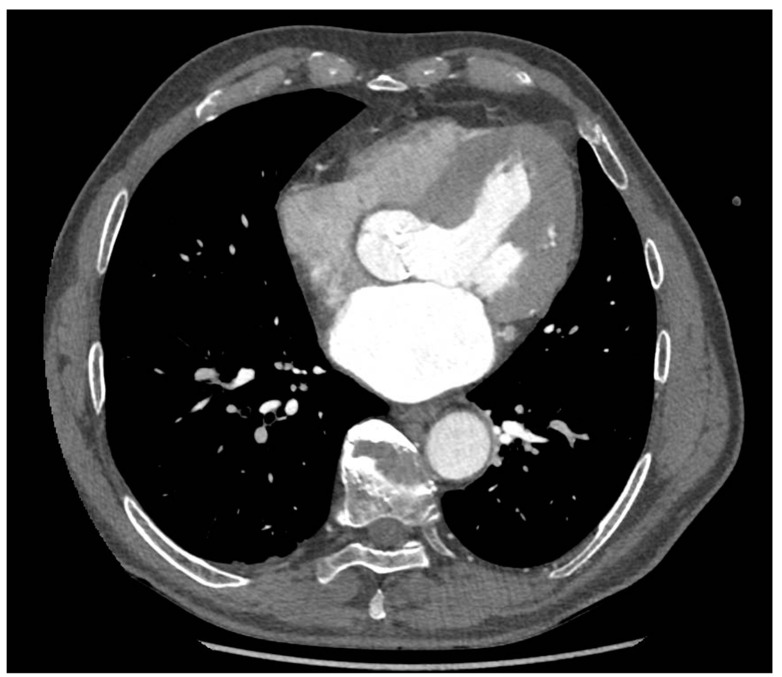
CT scan of the annulus pre-TAVI.

**Figure 2 jpm-13-00838-f002:**
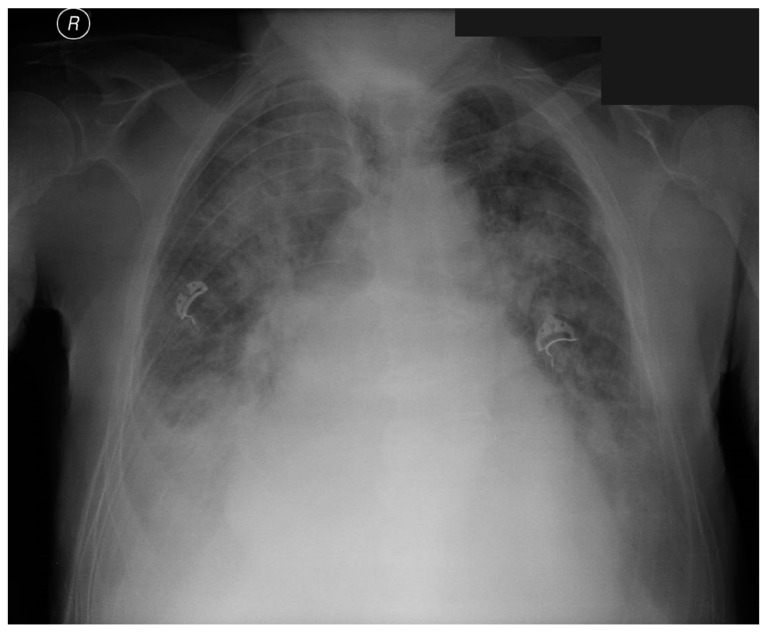
Chest X-ray on admission manifesting pulmonary edema.

**Figure 3 jpm-13-00838-f003:**
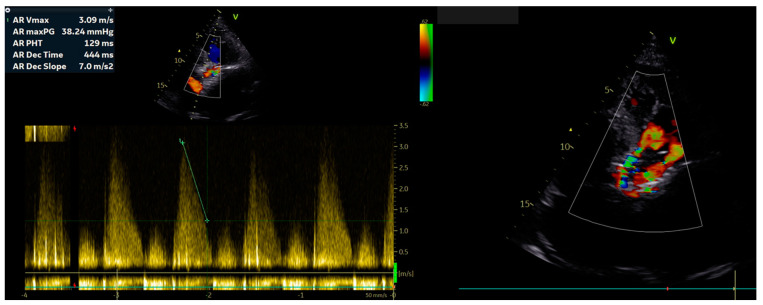
Post-TAVI ECHO depicting the remaining paravalvular aortic leakage.

**Figure 4 jpm-13-00838-f004:**
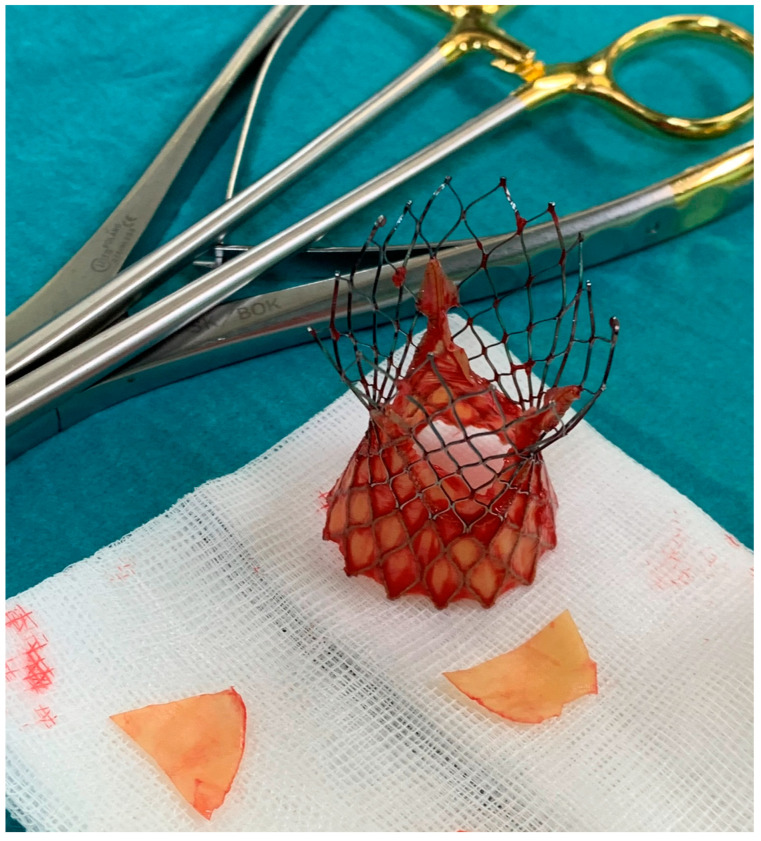
The explanted 26 mm Medtronic CoreValve Evolut Pro TAVI.

**Figure 5 jpm-13-00838-f005:**
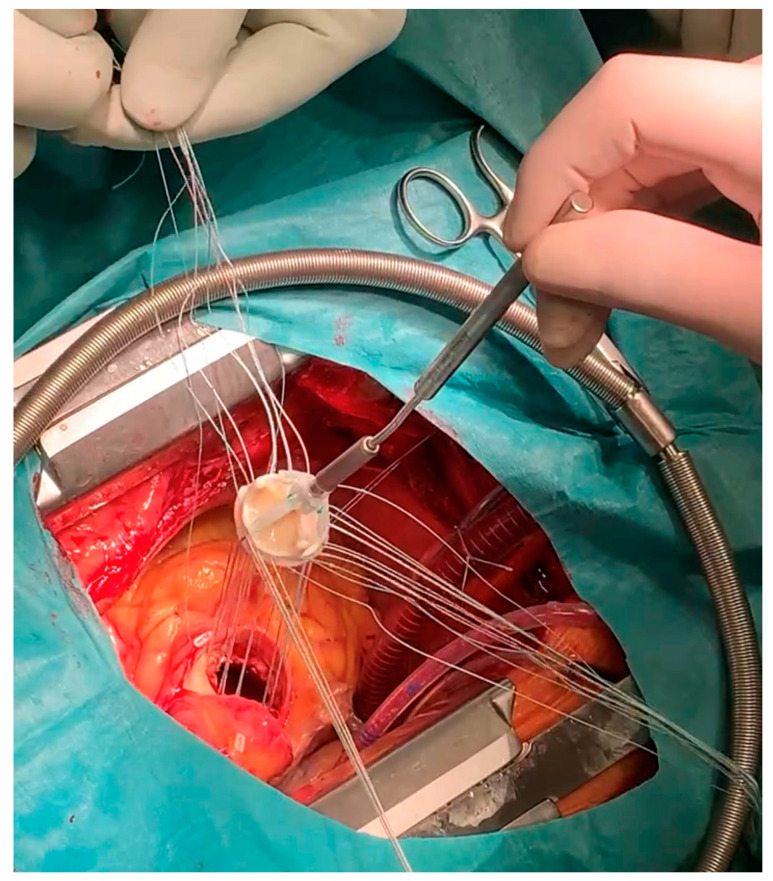
The implantation of biological prosthesis Edwards Perimount Magna size 25.

## Data Availability

The data presented in this study are available on request from the corresponding author. The data are not publicly available owing to privacy reasons.

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
