# Peer review of "Open-Heart Cardio-Thoracic Biological Valve Replacement Following Complicated Transcatheter Aortic Valve Implantation"

_jpm, 2023, doi:10.3390/jpm13050838_

Round 1

Reviewer 1 Report

If this article is considered a case report is too long; there is no necessity to describe the physiopathology of TAVI and it can shortened , in opposite if the Editor accept a review and the case report reinforces the concept of what discussed in the review , the paper is acceptable .

The quality of English is good

Author Response

Point 1: If this article is considered a case report is too long; there is no necessity to describe the physiopathology of TAVI and it can shortened, in opposite if the Editor accept a review and the case report reinforces the concept of what discussed in the review, the paper is acceptable”.

Response 1: Thank you, the case report has been shortened to the most crucial information. We also corrected the remaining English mistakes with the help of the professional. All the changes have been highlighted in red.

Reviewer 2 Report

Dear Editor and Authors,

Thank you for asking me to review this case report titled “Open-heart cardio-thoracic biological valve replacement following complicated Transcatheter Aortic Valve Implantation” by Dr. Klotzka and co-authors from the Departments of Cardiology and Cardiac Surgery at Poznan University of Medical Sciences in Poznan, Poland.

In this report the authors present the case of a 79 year old patient who developed a paravalvular aortic leak post transcatheter aortic valve implantation (TAVI) and who eventually had to undergo an open aortic valve implantation with a Perimount Magna valve.

This report I am afraid needs significant editing and rephrasing because not only is the case challenging as it is the way the authors report it make it look like they performed malpractice (which I don’t think they did but any other reader could think so!!). We need to present this case correctly and use it as an example of how not to do things!!

1.       The description of aortic stenosis *symptoms, presentation, management) in the introduction is superfluous and doesn’t add anything to the manuscript. We are all aware of the pathophysiology and management of AS so a review is not needed! The introduction needs to be reduced in size and should only focus on TAVI therapy and the potential complications of it. Also, the authors could talk more about patient selection for TAVI, how it is done, what is the risk profile of these patients, why is TAVI utilized ect. So only lines 105 to 117 are useful in the introduction. ALL THE REST (30 to 16) NEED DELETING!

2.       What do you mean the patient was admitted because of anxiety!!! Was he admitted under psychiatry? Anxiety is not a cardiovascular diagnosis, did the ER physicians not identify he had AS on their examination/work up?? With a AV PGmax of 85 his AS would be audible not with a stethoscope, just by ear they could hear it!! This whole section needs to be re-worked and re-written!

3.       What do you mean in 2022 the patient underwent TAVI? Was not TAVI performed following his initial admission? Again, the case presentation does not make sense, needs to be more clear and flow better!

4.       There is no shallow sedation as a term. It is called Monitored Anaesthetic Care!! I presume there was an anaesthesiologist assisting/providing MAC during the procedure!!

5.       Six re-positions for the valve seem excessive!! Why was that done? Was the anatomy difficult? Why did the valve slip (shifted), this seems like an error of inexperience!!

6.       You had a moderate perivalvular leak post procedure and this was acceptable?? You are wondering why the patient was re-admitted a couple of days later with pulmonary oedema!!

7.       What was the patients EUROSCORE? Was he discussed in the multidisciplinary heart team so that he could be referred for a TAVI?

8.       Was a CT aortogram performed so as to identify the tortuosity of the aorta and the difficult anatomy?

9.       What was the experience of the unit when this case was performed? Where they at the beginning of their TAVI experience?

10.   The description of the cardiac operation needs to be better formatted, it seems like a medic/cardiologist is describing the procedure instead of a surgeon. A surgeon like me would say “Under femoral – SVC / IVC cardiopulmonary bypass (CPB) a median sternotomy was performed. Despite good….observed. The pericardium was opened and following cardioplegic arrest of the heart and aortic cross clamping an aortotomy was performed. The implanted stent valve was removed……” something like this!!

11.   No follow up is reported about the patient after he was transferred to cardiac ICU. What happened afterwards, was he extubated early, was he discharged home, how was he in follow up, was he back to normal activities or symptomatic, DID HE DIE???

12.   If the patient could undergo an emergent open aortic valve implantation why did he not undergo this to begin with??

In conclusion, this is NOT a ground breaking / rare case!!! It is something that can happen and has happened in all operators of TAVI. The main point is what to do to avoid it happening (this needs to be included a bit more in the discussion) and how to correct it!! I am very close to rejecting this work but I feel that it needs to be presented in the literature as a cautionary tale!! HOWEVER, IT NEEDS MAJOR REVISION SO…  

Needs some work to improve both expressions and minor spelling mistakes.

Round 2

Reviewer 2 Report

Dear Editor and Authors,

Following a particularly extensive and exhaustive review I am happy to be asked again to re-evaluate this revised manuscript. It was good to see that the authors have heeded my comments and have made the requested corrections. As such the case report is I feel now much clearer and improved.

However, there still some small inconsistencies that need some minor changes/editing to make it clearer prior to this manuscript been ready for publication. Mind you, and so as to not disappoint the authors it is quite good but in two places (were the post-operative course of the patient and his discharged is described it needs some correction). Specifically, in lines 158 to 161 the authors describe the patient’s course but it is a bit puzzling. They mention the patient was extubated after the TAVI procedure but according to what they previously described he was not intubated and anesthetized but only under MAC and local anesthetic!! Also, again in lines 165 to 169 the authors describe the patient’s post-TAVI course and mention his discharge home. Please clarify this section better.

Similarly, following the surgical valve replacement. We usually do not extubate patients right after cardiac surgery is completed (this is something the authors would not know because they are cardiologists but there are 4 cardiac surgeons as co-authors so this should not have passed them by!!) but instead they get transferred to cardiothoracic ICU where slowly they are re-warmed, electrolyte balances are restored, bleeding is assessed, and gradually they are weaned off sedation and extubated!! You need to mention this process and please, please get the cardiac surgeons to help you. They need to go over the surgical procedure again and refine it. We do not say “performed the middle longitudinal sternotomy” we say a midline sternotomy was performed!! Why did you do femoral cannulation and did not cannulate the aorta?

Why did you do bicaval canunnation for an aortic valve replacement, did you open the atrium? It is just that the section reads like it was not written by a surgeon, for example we do not “do an aortic incision” but an aortotomy was performed!! Just the lingo doesn’t jibe!!

So, some minor corrections again (much better review this time I feel compared to the big list of comments previously), get the surgeons to help you a bit on the write up and will accept the submission no problem. All the best.

Needs some extensive language and expression editing.
